# Unmasking the Web of Deceit: Uncovering Coordinated Activity to Expose Information Operations on Twitter

## ABSTRACT

Social media platforms, particularly Twitter, have become pivotal arenas for influence campaigns, often orchestrated by state-sponsored information operations (IOs). This paper delves into the detection of key players driving IOs by employing similarity graphs constructed from behavioral pattern data. We unveil that well-known, yet underutilized network properties can help accurately identify coordinated IO drivers. Drawing from a comprehensive dataset of 49 million tweets from six countries, which includes multiple verified IOs, our study reveals that traditional network filtering techniques do not consistently pinpoint IO drivers across campaigns. We first propose a framework based on node pruning that emerges superior, particularly when combining multiple behavioral indicators across different networks. Then, we introduce a supervised machine learning model that harnesses a vector representation of the fused similarity network. This model, which boasts a precision exceeding 0.95, adeptly classifies IO drivers on a global scale and reliably forecasts their temporal engagements. Our findings are crucial in the fight against deceptive influence campaigns on social media, helping us better understand and detect them.

### ACM Reference Format:
Anonymous Author(s). 2024. Unmasking the Web of Deceit: Uncovering Coordinated Activity to Expose Information Operations on Twitter. In *WebConf '24: ACM Web Conference, May 13–17, 2024, Singapore.* ACM, New York, NY, USA, 11 pages. https://doi.org/XXXXXXX.XXXXXXX

## 1 INTRODUCTION

Social media have become a fertile ground for the orchestration and execution of influence campaigns. These manipulative efforts are designed to shape public perception by disseminating fabricated and deceptive information, typically to promote a specific political viewpoint or ideology. Such initiatives are most prevalent during pivotal geopolitical events [37, 41], such as elections or crises, where the drivers of these campaigns exploit the naturally-occurring online chatter to spread politically biased content, sow division among opposing factions, or target influential users [42, 47, 52]. Among their possible forms, influence campaigns can take the shape of state-sponsored information operations (IOs), wherein government-backed actors collaboratively disseminate propaganda and misinformation aligned with their own ideologies or aimed at undermining opposing viewpoints.

A substantial body of research links orchestrated campaigns by state-sponsored entities to attempts at manipulating public opinion on social networks during pivotal political events [3, 16, 43]. The 2016 U.S. Presidential Election, targeted by Russian IO, exemplifies this, with bots and trolls disseminating content on social media platforms [4, 57]. Similarly, IOs by the Chinese Communist Party (CCP) allegedly use coordinated actors on social media to influence public opinion [23, 24].

An influence operation's life cycle[1] involves three steps. Initially, operations create fake and automated accounts to mimic genuine users [13, 27, 33]. These personas then generate and spread content, often in coordination [30, 39, 54]. Organic users might engage with this content, amplifying its reach, sometimes even to mainstream media [9, 28, 29]. This study zeroes in on the second step, avoiding the identification of independent inauthentic personas or modeling organic user susceptibility. IOs are typically coordinated efforts by multiple inauthentic users [38, 40, 46]. We term this group *IO drivers*, following [38]. These actors use various techniques, including artificially boosting content, manipulating platform feeds, and engaging key users [36, 37, 47].

Extensive research aims to detect online coordination by identifying unexpected similarities in user actions [5, 32, 36, 39, 40, 54]. These similarities span behaviors like co-retweeting and synchronized posting. Such patterns form the foundation for networks that depict user similarities using edge weights. The premise is that connections between similar users can unveil coordinated user clusters. To improve accuracy in identifying orchestrated campaign accounts and minimize organic user misclassification, current methods filter low-weight edges in similarity networks by setting high similarity thresholds. This choice is also driven by the absence of ground truth in previous studies.

### Contributions of this work

With the release of datasets on Twitter IOs [18], this paper evaluates existing methods, investigates new cues to detect coordinated actions, and introduces novel techniques to identify influence campaigns from multiple countries. We aim to surpass known filtering approaches by leveraging topological features and properties of similarity networks, like node embedding and centrality, relying upon five behavioral traces to build similarity networks. The paper addresses the following three Research Questions (RQs):

**RQ1**: *To what extent can known edge-weight filtering approaches identify IO drivers? Is there a specific behavioral trace that consistently enables IO drivers' detection for every IO?*: We demonstrate that edge-weight filtering approaches exhibit limited capabilities in consistently detecting IO, even when their parameters are optimized, highlighting the need for alternative approaches to advance the state of the art.

---

[1]We use *Influence Operation* and *Information Operation* interchangeably.

**RQ2**: *Does centrality-based node pruning yield better classification performance compared to edge filtering approaches? Does combining network similarities result in improved classification performance?*: We show that node pruning surpasses edge-weight filtering across different IOs and behavioral traces, demonstrating how node centrality signals IO drivers more accurately than edge weights. Nevertheless, our analysis underscores the necessity of solutions that can integrate various behavioral traces to detect diverse IOs. We provide evidence of the beneficial impact of combining siloed similarity networks in a unique network that accurately identifies coordinated actors based only on their centrality in this fused network (AUC = 0.84, F1 = 0.77).

**RQ3**: *Can similarity networks' embeddings enable the detection of coordinated accounts across multiple interacting influence campaigns? Can these network representations be used to predict users' involvement in an IO?*: By generating a vector representation of the fused similarity network, we introduce a supervised machine learning approach capable of detecting IO drivers across various campaigns using only behavioral traces (AUC = 0.95, F1 = 0.83). This approach was also tested in challenging scenarios, wherein our conservative model successfully classifies IO drivers on a global scale and accurately predicts their involvement over time with a precision exceeding 0.95.

Using a data set comprising 49M tweets from the Twitter Information Operations archive [18], this article performs an analysis of influence campaigns that originated in six different countries. Our study evaluates existing methods and proposes novel computational models to identify coordinated networks of IO drivers. Overall, we provide foundational insights and novel directions to research endeavors focused on harnessing behavioral trace similarities to uncover coordination within influence campaigns.

## 2 RELATED WORK

IO detection has been approached from various perspectives: either by analyzing individual inauthentic users or by examining the collective behavior of malicious account networks.

### 2.1 State-sponsored IOs and their identification

Research has extensively analyzed individual account activities to detect participation in influence campaigns, focusing on entities such as bots (software-controlled accounts) and trolls (state-backed human operators) [14, 33].

For bots, solutions have used various features and machine learning strategies to identify bot characteristics [8, 10, 56]. Botometer [55, 56] has been instrumental in scaling bot activity research on Twitter. However, recent studies emphasize that IO coordination isn't solely automated [21, 36].

Research on state-sponsored trolls has been categorized into three categories based on detection features: content-based methods [1, 2, 22], behavioral-based approaches [26, 31, 45], and sequence-based techniques [12, 38]. Unlike these methods, our paper focuses on group-level coordination, emphasizing orchestrated campaigns over isolated inauthentic efforts.

### 2.2 Coordination Detection

Automated detection of coordinated IOs has employed various strategies. Temporal methods, like the *Rapid Retweet Network* approach [39, 47], focus on synchronized posting times as indicators of suspicious activities [6, 7, 32, 39, 40, 48].

Content-based techniques, such as the *Tweet Similarity* [39, 47] and *Hashtag Sequence* methods [5], analyze shared content among users. Others focus on shared URLs [17] or news articles [19].

Interaction-based methods, like the *Co-Retweet* [36, 40], examine user interactions such as retweets and mentions. State-of-the-art methods explore latent coordination signals [11, 35, 49, 51, 54]: For instance, Vargas et al. [51] use time-series analysis, while Sharma et al. [45] focus on mutual influence leading to collective behavior.

Our approach differs from existing methods, which primarily construct similarity networks based on a single behavioral trace. We harness the topological properties of the similarity network, emphasizing node centrality and embedding. We aim to capture coordinated actors across a broad IO spectrum by evaluating diverse user similarities and their combinations.

## 3 DATA

In our quest to uncover coordinated actions behind influence campaigns, we center our analysis on IOs on Twitter. The platform has suspended accounts associated with these operations for violating their terms of service, which describe platform manipulation as attempts to artificially amplify conversations using tactics like multiple accounts, fake accounts, and automation.[2]

To foster transparency and research, Twitter has shared over 141 IO datasets from 21 countries, detailing every tweet from each IO driver since account inception.

*IO campaign data.* Our analysis focuses onto six countries: China, Cuba, Egypt & UAE, Iran, Russia, and Venezuela. These countries were selected based on the extensive scale of their IOs, evident from their vast user base. In line with recent studies [26, 53], we examine IOs at the country level, combining campaigns from the same country, as outlined in Table 1. This approach mirrors real-world situations where multiple campaigns and organic conversations from a single country might intersect. Notably, based on Twitter's insights [18] and prior research [53], we've combined accounts linked to both Egypt and the UAE, as their IOs predominantly targeted Iran and Qatar.

*Control data.* For a comprehensive evaluation of coordination detection methods, we need a control group of organic users. We employ the dataset by Nwala et al. (2023) [38], comprising tweets from genuine users discussing similar topics in the same time frames as the IO drivers. This dataset was curated by extracting hashtags from IO drivers and querying them in Twitter's academic search API. Results were filtered to pinpoint accounts active during the IO drivers' active periods, and up to 100 tweets from these control users during the respective IO were compiled.

## 4 METHODS

This section delves into both existing and proposed methodologies for detecting coordinated activities in IOs. We begin by elucidating

---

[2]https://help.twitter.com/en/rules-and-policies/platform-manipulation

| Country (no. of campaigns) | Accounts Lifespan | IO Drivers [tweets] | Control Users [tweets] |
|---|---|---|---|
| China (1) | 2010-2019 | 5,191 [13.8M] | 76,286 [3.5M] |
| Cuba (1) | 2010-2020 | 503 [4.8M] | 30,099 [1.4M] |
| Egypt & UAE (2) | 2011-2019 | 240 [1.5M] | 370 [0.4M] |
| Iran (5) | 2010-2020 | 209 [9.9M] | 16,885 [2.5M] |
| Russia (5) | 2010-2020 | 3,487 [9.8M] | 31,317 [4.4M] |
| Venezuela (2) | 2010-2019 | 33 [9.5M] | 3,865 [0.7M] |

Table 1: IOs examined in this work. For each IO, we report their accounts' life span, the number of IO drivers, control users, and their corresponding volume of tweets

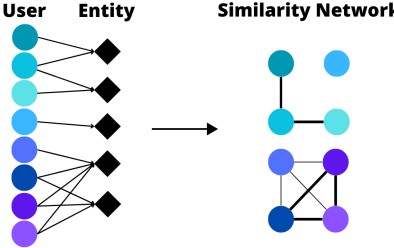

**Figure 1: Construction of similarity graphs from behavioral traces**

the foundational assumptions and strategies for constructing similarity networks from various behavioral traces. Subsequently, we detail the techniques we have developed, rooted in these similarity networks, and their potential applications.

## 4.1 Framework Overview

At the core of coordination detection methods lies the assumption that genuine users operate independently, exhibiting limited similarities in their online behaviors [39]. Thus, any unexpected convergence in behavior can hint at potential coordination among users [40]. Building on this assumption, existing techniques harness user activity features, termed here as *behavioral traces*, to gauge similarity between users. In our study, we incorporate five distinct behavioral traces, including sharing identical links, hashtags, or content, re-sharing the same tweets, or exhibiting automation-enabled actions such as rapid retweeting [34, 39].

These coordinated behaviors are often tactics in IOs, aiming to artificially boost content, fabricate a sense of consensus, or manipulate platform algorithms [15, 40, 47]. Each behavioral trace (§4.2) helps to create a similarity network (§4.3), where user similarities are represented through edge weights. Using these networks, we

identify coordinated groups via three methods: (*i*) a popular unsupervised technique based on edge filtering (§4.4.1); (*ii*) our novel unsupervised approach centered on node pruning (§4.4.2); and (*iii*) a new proposed supervised strategy rooted in graph embedding (§4.6).

## 4.2 Behavioral Traces

Next, we delineate the behavioral traces used in our study, then outline the process of creating each corresponding similarity graph. We have identified five primary behavioral traces:

- *Co-Retweet*: The act of re-sharing identical tweets.
- *Co-URL*: Disseminating the same link or URL.
- *Hashtag Sequence*: Using an identical sequence of hashtags within tweets.
- *Fast Retweet*: Quickly re-sharing content from the same users.
- *Text Similarity*: Posting tweets with closely resembling textual content.

While this list captures the primary traces we have focused on, it is by no means exhaustive. Other potential similarities, such as temporal patterns and synchronized posting times [40], were assessed. However, they were excluded from our framework due to their limited effectiveness in pinpointing coordinated IO drivers. In the future, we will operationalize and assess additional behavioral traces associated with IOs.

## 4.3 Constructing Similarity Graphs

The process of creating a similarity graph is largely consistent across most behavioral traces, as illustrated in Figure 1. We start by forming a bipartite graph between users and entities, the latter representing the specific behavioral trace under consideration (e.g., for the Co-URL trace, entities are the URLs). This bipartite network links users to entities based on their sharing activities, with weights assigned using TF-IDF to reflect the popularity of each entity. Consequently, each user is depicted as a TF-IDF vector of the shared entity. This bipartite graph is subsequently transformed into a similarity network, connecting users based on their behavioral trace similarities. The connections are weighted, with the weight determined by the cosine similarity between the TF-IDF vectors.

For the *Co-Retweet*, *Co-URL*, and *Hashtag Sequence* traces, the construction process is analogous but utilizes distinct inputs. For the Co-Retweet network, a bipartite graph is formed between users and tweets, linked by retweet activity. For Co-URL, URLs within tweets are extracted to form a bipartite graph. The Hashtag Sequence trace employs an ordered sequence of hashtags, with an added parameter to set the minimum number of hashtags in a sequence. The *Fast Retweet* network focuses on rapidly repeated retweets, using a time threshold to classify a retweet as "fast". From this refined set, a bipartite network is constructed, which is then weighted using TF-IDF based on the popularity of each targeted user, and subsequently projected onto a similarity network.

The *Text Similarity* trace diverges from the above strategy. Instead of a bipartite graph, a direct similarity network is formed, weighted by the cosine similarity of users' shared textual content. This content, excluding retweets, undergoes a cleaning process to remove punctuation, stopwords, emojis, and URLs. Only tweets

with a minimum of four words are considered, as shorter texts were found to be less relevant and risked introducing noise. We employ the Sentence Transformer *stsb-xlm-r-multilingual* from Hugging Face for text embeddings, calculating cosine similarity using the efficient FAISS algorithm [25]. To optimize computational efficiency, we assess similarities within a one-year sliding window. A similarity threshold, set at 0.7 according to previous research [39, 47], ensures that only tweets that are the most similar are considered. The resulting *Text Similarity* network connects users if they post at least one pair of similar tweets, with the average text similarity serving as the edge weight.

## 4.4 Unsupervised Coordination Detection through Network Dismantling

This section elucidates unsupervised methodologies that utilize the inherent properties of similarity networks to identify coordinated IO drivers. We delve into two primary strategies: edge filtering and node pruning.

*4.4.1 Low-weight Edge Filtering.* Edge filtering is a predominant technique in detecting coordinated activities [5, 39, 40, 47]. It operates on the premise that the strength of similarity between users can spotlight coordinated entities. In this context, the weight of an edge in a similarity network signifies the strength of similarity between two users. By setting a similarity threshold, prior research has filtered out weaker connections to reveal clusters of coordinated users. Notably, users who remain unconnected post-filtering aren't deemed coordinated. Given the absence of ground truth in many studies, a conservative threshold has traditionally been used to exclude potentially independent users. In our study, we evaluated this method on different IOs, both by adhering to this conservative threshold and by optimizing it to improve detection accuracy (§5.1).

*4.4.2 Network Pruning based on Node Centrality.* We introduce a novel strategy that emphasizes node pruning in similarity networks based on centrality measures. The fundamental idea is that IOs, involving multiple accounts, often manifest a pronounced collective similarity. In a similarity network, this is evident when a node (representing an IO driver) connects to numerous other nodes. As illustrated in Figure 2A, IO drivers typically occupy central positions in the similarity network, while organic users are more peripheral. Panels B and C of Figure 2 further differentiate IO drivers from organic users based on edge weight and node centrality, respectively. While edge weight distributions reveal discernible differences between the two user types, node centrality seems even more potent in distinguishing them.

Our analysis leverages eigenvector centrality, which has demonstrated superior discriminative power compared to other centrality measures. A comprehensive comparison is available in Appendix Fig. 10. For nodes absent in certain similarity networks, a centrality value of 0 is assigned. After computing centralities, nodes with lower eigenvector centrality are pruned. Like edge filtering, we evaluated this method in different IOs, presenting results with optimized and conservative centrality thresholds to pinpoint coordinated actors (§5.2).

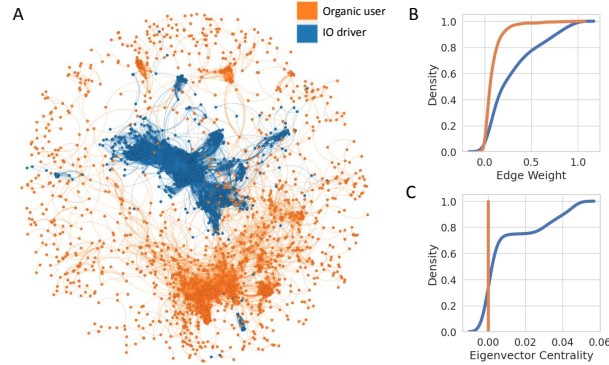

**Figure 2: Panel A: Co-Retweet similarity network of users from Egypt & UAE. Blue nodes indicate IO drivers, whereas orange nodes represent organic users. Panel B and C depict the CDF of edge weight and eigenvector centrality of the two classes of accounts, respectively**

## 4.5 Network Fusion for Enhanced Similarity Detection

Traditional methods often analyze a single similarity network or a limited subset in isolation. However, we posit that relying solely on one behavioral trace might not comprehensively identify all IO drivers. This is grounded in the understanding that individual accounts might employ a diverse array of strategies, leading to different user groups orchestrating varied coordinated actions.

To address this, we introduce the concept of a *Fused Network*, which combines multiple similarity networks, encompassing Fast Retweet, Text Similarity, Co-Retweet, Co-URL, and Hashtag Sequence. This fusion aims to enhance the detection accuracy and generalizability by capturing a broader range of coordinated behaviors.

In our exploration of the fusion process, we assess various strategies for integrating these networks, applicable to both edge-filtering and node-pruning methods. These strategies range from aggregating normalized weights to choosing the maximum centrality of individual similarity networks. The most effective approach we found links two nodes in the *Fused Network* if they are connected in any of the individual similarity networks. Although there are many other possible fusion strategies, our focus remains on underscoring the advantages of amalgamating multiple similarity metrics to enhance the detection of coordinated IO activities (§5.2).

## 4.6 Supervised Detection Using Coordination Signatures

While Section 4.4 explored unsupervised techniques suitable for contexts without ground truth, this section focuses on supervised models. These models leverage labeled data to craft classifiers that pinpoint IO drivers using coordination indicators. Given the rich information embedded in the similarity networks, our supervised approach seeks to harness their topological nuances, both individually and in a combined fashion.

| Behavioral Trace (parameters, pt) | Prior Work (AUC) (pt) | Optimized (AUC) percentile (pt) |
|---|---|---|
| Fast Retweet (time interval) | 0.53 ± 0.03 (10s) | 0.62 ± 0.13 50-th (60s) |
| Co-Retweet (percentile) | 0.55 ± 0.03 (99.5-th) | 0.69 ± 0.09 80-th |
| Co-URL (percentile) | 0.61 ± 0.04 (99.5-th) | 0.72 ± 0.09 80-th |
| Hashtag Sequence (no. hashtags) | 0.59 ± 0.07 (5) | 0.68 ± 0.16 65-th (3) |
| Text Similarity (cosine similarity) | 0.47 ± 0.04 (0.7) | 0.52 ± 0.05 96-th (0.95) |

**Table 2: Average AUC of prior work when using their parameters (pt) vs. optimized ones**

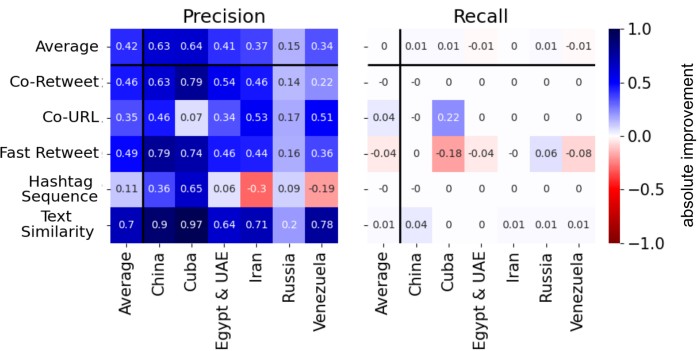

**Figure 3: Improvement in the classification performance by using node pruning instead of edge filtering**

However, directly applying machine learning to network structures poses challenges. To overcome this, we employ node embeddings, specifically using Node2Vec [20]. This technique translates the network's structure into a more digestible, low-dimensional space, producing vector representations of length 128. For each node, we initiate 16 walks, each spanning 16 steps, to derive its embedding. With these embeddings in hand, we deploy standard machine learning algorithms for several classification tasks:

**Task 1**: Distinguishing users involved in separate IOs, utilizing both individual and fused similarity networks.

**Task 2**: Classifying users on a global scale, accounting for potential overlaps and similarities among multiple IOs.

**Task 3**: Forecasting user participation in IOs over varying years of activity.

These tasks underscore the potential of supervised models that rely on representations derived from the similarity network. They are particularly relevant in real-world scenarios (§5.3) where social media platforms release annotations sporadically and IOs can intersect with a mix of genuine and coordinated discussions.

## 5 EVALUATION

In this section, we delve into the performance metrics of our unsupervised methods for detecting coordinated activity, specifically focusing on edge filtering and node pruning. Subsequently, we shift our attention to the results from the supervised embedding-based model across the three previously outlined classification tasks. Our evaluation metrics encompass Precision, Recall, F1, and AUC.

### 5.1 Assessing IO Detection via Edge Filtering (RQ1)

In RQ1, we examine the efficacy of common edge filtering techniques in detecting a variety of IOs. Our analysis adopts parameters established in prior studies: for *Co-Retweet* and *Co-URL*, the 99.5-th percentile of cosine similarity in co-sharing activities [36, 40]; for *Hashtag Sequence*, a minimum sequence of 5 hashtags [40]; for *Fast Retweet*, a 10-second window [39]; for *Text similarity*, a cosine threshold of 0.7 [39, 47].

Next, we refine these parameters to optimize the AUC classification performance. We also employ a TF-IDF-weighted bipartite graph framework across all behavioral traces, ensuring a uniform metric of cosine similarity. In line with prior research, we individually assess the five behavioral traces and their associated similarity networks.

Table 2 contrasts the classification performance (AUC) between the prior work and our optimized parameters. The parameters from prior work [36, 39, 40, 47], appear to be more stringent than our optimized set on most behavioral traces, with the exception of text similarity. This cautious approach likely stems from a desire to reduce false positives in contexts without a clear ground truth. Importantly, our study is the first to evaluate edge filtering techniques in a context where IO annotations are available.

Furthermore, even post-optimization, the AUC performance exhibits considerable variability, ranging from 0.52 to 0.72 depending on the behavioral trace. This disparity underscores that, while a particular behavioral trace might be adept at detecting certain IOs, it might falter with others. A deeper dive into the performance metrics across various IOs and countries reinforces this observation. For a more granular breakdown, the reader is directed to Fig. 12 in the *Appendix*.

*Key Insights.* The edge filtering technique, particularly the low-weight variant, demonstrates inconsistent efficacy in detecting a wide range of IOs. Even with parameter optimization for each similarity network, the method showcases potential in pinpointing actors in specific influence campaigns, but struggles to maintain this accuracy universally across all IOs.

### 5.2 Comparative Analysis: Node Pruning vs. Edge Filtering (RQ2)

Historically, research has emphasized edge weights to detect coordinated IO drivers, based on the strength of similarity between users. However, this approach may miss out on capturing the broader behavioral similarities users might exhibit along different axes, especially smaller ones, if taken individually. To address this, we introduce node centrality within a similarity network as a more encompassing measure.

Initially, we evaluated our node pruning approach against the traditional edge filtering method. For a fair comparison, both models are optimized for their best parameters, to maximize precision

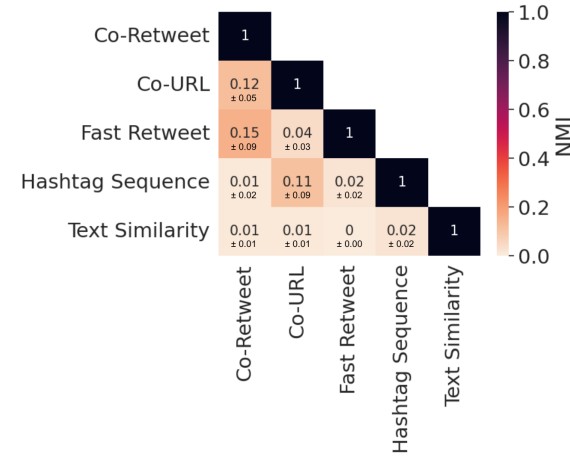

**Figure 4: Average NMI score between groups of users involved in different coordinated actions**

(resp., recall), shown in the left (resp., right) panel of Fig. 3. We observe comparative improvements in precision and recall when transitioning from edge filtering to node pruning. Our findings indicate that node pruning enhances precision by an average of 0.42 while maintaining comparable recall levels. This improvement is consistent across various campaigns and similarity networks, with only a few deviations. This is a rather important feature of our model, since misclassifying organic users has a higher cost, resulting in potential penalties (e.g., account suspension) to innocent users.

A deeper dive into performance metrics across diverse IOs and similarity networks confirms that no single behavioral trace consistently captures different IOs across all countries. For example, while the Co-URL similarity graph effectively identifies Chinese, Russian, and Venezuelan coordinated accounts, it struggles with IOs from Cuba and Iran. Moreover, different groups of users within an IO may employ a diverse suite of strategies. This observation is confirmed in Figure 4, which illustrates the Normalized Mutual Information (NMI) score between groups of users engaged in various coordinated actions. NMI scores close to zero indicate minimal overlap between groups. As a result, a particular similarity network can only identify a subset of users within the IO drivers' spectrum (see Table 3). This variability suggests that IO campaigns employ a diverse range of tactics and that a single similarity network might only capture a subset of these coordinated actions.

To address this limitation, we introduce a fused similarity network that combines multiple behavioral traces. This fusion, as illustrated in Figure 5, enhances the generalizability of the model across various campaigns. The fused approach does not necessarily improve the classification performance for each campaign, but ensures consistent accuracy across various IOs.

In summary, our fused network approach achieves an average AUC of 0.83 and an F1 of 0.76. Notably, these results are based on an unweighted version of the eigenvector centrality. When weights of the fused similarity network are considered for computing node centrality, the classification performance does not improve. Similarly, various combinations of edge filtering and node centrality,

| Country | Proportion of IO Drivers | | | | | |
| | FR | CR | CU | HS | TS | Fused |
|---|---|---|---|---|---|---|
| Egypt & UAE | 11% | 76% | 89% | 70% | 81% | **96%** |
| Cuba | 71% | 94% | 44% | 82% | 73% | **97%** |
| Iran | 22% | 62% | 61% | 25% | 64% | **87%** |
| Russia | 34% | 59% | 94% | 60% | 90% | **97%** |
| China | 12% | 58% | 77% | 17% | 32% | **84%** |
| Venezuela | 61% | 85% | 91% | 38% | 77% | **96%** |

**Table 3: Proportion of IO drivers captured by each similarity network in IOs from Egypt & UAE, Cuba, Iran, Russia, China, and Venezuela. FR = Fast Retweet, CR = Co-Retweet, CU = Co-URL, HS = Hashtag Sequence, TS = Text Similarity**

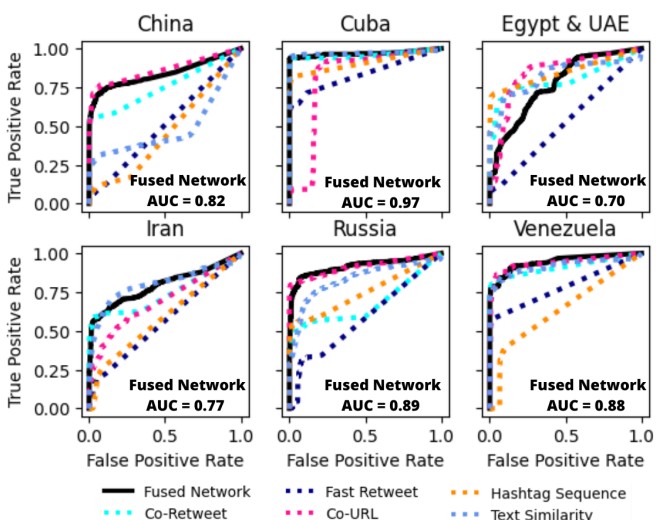

**Figure 5: AUC ROC of siloed and fused similarity networks**

or alternative approaches based on multiscale filtering methods [44] as suggested in [36, 48], do not appear to offer substantial improvements in predictive performance and may even result in performance degradation (see Table 4 in the *Appendix*). Exploring this further is earmarked for future research.

*Key Insights.* Our node pruning methodology demonstrates superior performance over traditional edge filtering techniques in identifying coordinated IO drivers. The results emphasize the need for a holistic approach, combining multiple behavioral traces, to capture the various tactics employed by IO campaigns. This method can be applied unsupervised when ground truth data is unavailable. For optimal results, we advocate for the fusion of multiple similarity networks and recommend a conservative centrality threshold.[3]

---

[3]Based on our experiments, a centrality threshold of $10^{-2}$ ensures a Precision > 99%, while maintaining an average AUC > 70%.

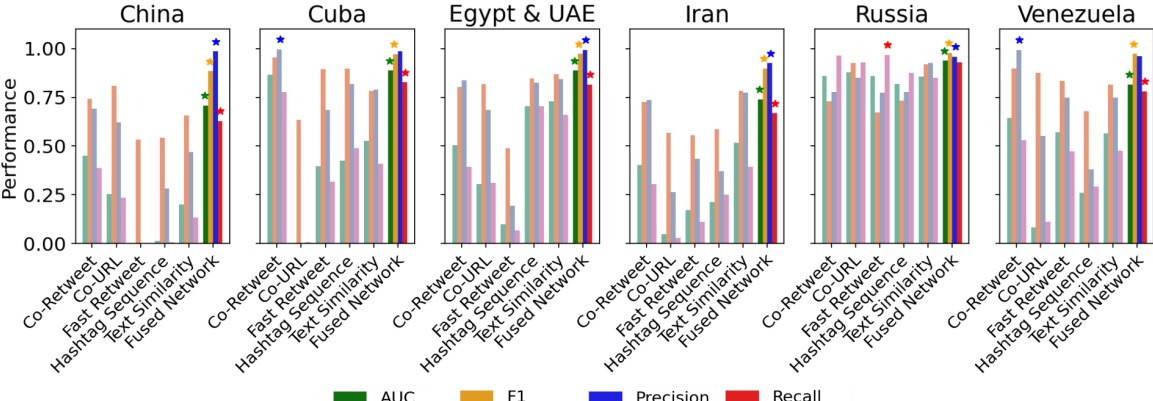

Figure 6: Classification performance of the siloed and fused similarity networks

## 5.3 Harnessing Embeddings from Similarity Networks for Classification (RQ3)

To address RQ3, we transition from raw similarity networks to a more compact representation using node embeddings, converting users into 128-dimensional vectors. This transformation aims to encapsulate the intricate topological structure of the similarity network, thereby facilitating our three classification tasks (cf., §4.6).

*5.3.1 Task 1: IO Drivers Detection.* The potential of our embedding approach is visually captured in Figure 7, where the node embeddings from the fused similarity network are projected into a 2D space using t-SNE [50]. A clear demarcation between IO users and organic ones is evident, underscoring the method's efficacy.

We employ this approach on both siloed and fused similarity networks, using a Random Forest classifier with a 10-fold cross-validation to ensure the robustness of our results. Figure 6 presents the classification metrics, with the fused network approach consistently outperforming individual networks. On average, the fused approach achieves an AUC of 0.94, an F1-score of 0.82, and a remarkable precision of 0.96.

An ablation study further elucidates the contribution of each similarity graph within the fused network. While each trace adds value, the co-Retweet and Fast Retweet networks emerge as the most and least influential, respectively (see Appendix Fig. 13).

*5.3.2 Task 2: Classification on a Global Scale.* Broadening our scope, we combine interactions and similarities from all IO drivers into a unified fused similarity network. Figure 8 visualizes this *global* embedding space, revealing distinct clusters based on countries and potential inter-state collaborations. Temporal patterns also emerge, hinting at the longevity and strategy of different IO campaigns.

For classification, we replicate the Task 1 methodology but on this global fused network. The results are encouraging, with a precision of 0.95, recall of 0.70, F1-score of 0.78, and AUC of 0.92.

*5.3.3 Task 3: Forecasting Users' Engagement in IOs.* Finally, we assess the predictive capabilities of our approach. Using data from prior years, we aim to predict users who will engage in IOs in subsequent years. This evaluation is set in the global context of Task 2, adding another layer of complexity.

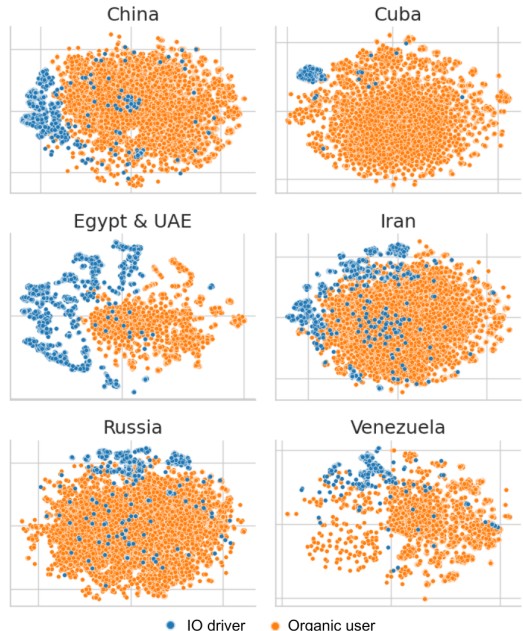

Figure 7: t-SNE visualization of node embeddings of the fused similarity network for different IOs

The results, presented in Figure 9, indicate a steady improvement in classification performance as more data become available. Notably, our model consistently achieves near-perfect precision and an F1 score exceeding 0.70 by 2017. This is a particularly significant result, considering that a substantial proportion of IO drivers became active in 2018 and 2019 (see Fig. 14 in the *Appendix*).

*Key Insights.* The embeddings derived from the fused similarity network prove instrumental in detecting IO drivers on a global scale and forecasting their future engagements. This supervised technique is best suited for scenarios where some ground truth data is available. For optimal results, we advise amalgamating various similarity networks to ensure a high-precision model.

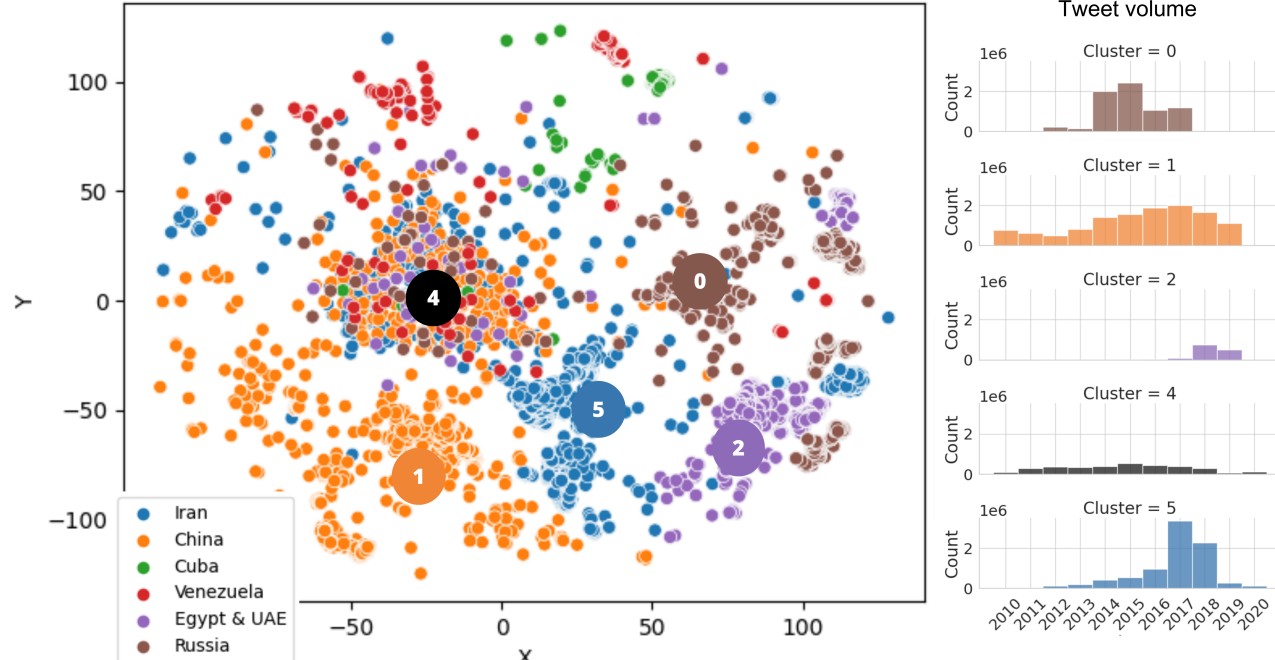

**Figure 8: t-SNE of *global* node embeddings for IO drivers**

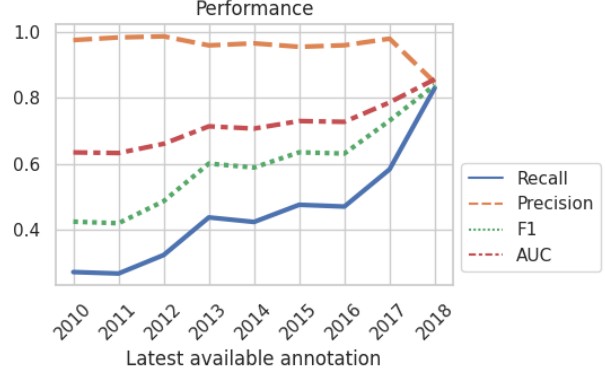

**Figure 9: Performance of model trained on historical data. We classify users who will engage in an IO after a specific year based solely on the users active in the preceding years**

## 6 CONCLUSIONS

In this paper, we introduce novel models for identifying coordinated actors driving IOs. Our approach proposes a paradigm shift from conventional coordination detection techniques. By prioritizing network properties, such as node centrality, we emphasize the detection of accounts that exhibit similarities with many others (node centrality). This diverges from earlier methods that focused on accounts highly similar to at least one other (edge weight). This shift in perspective allows us to leverage even weak similarity signals, resulting in more precise IO drivers' identification (42% improvement). Recognizing the need for a comprehensive approach that can generalize across campaigns from diverse countries, we propose the fusion of multiple behavioral indicators. Through a vector representation of a network combining five similarity traces, we propose a supervised approach that accurately distinguishes organic users from IO drivers in complex scenarios where diverse campaigns are intertwined. Our findings pave the way for novel methods that utilize user similarities to expose IOs, setting the stage for future research on the detection of state-backed IOs.

*Limitations.* Our work, while promising, has limitations. First, our definition of IO drivers is based on users identified by Twitter, but the exact mechanisms Twitter used for this identification remain opaque. Potential biases in data collection and possible misclassification of accounts can impact the detection efficacy of our models. Second, the activities or keywords used by control users might differ in frequency from those of IO drivers. These differences could imply that control users inherently constitute separate networks, not solely because of their non-IO status. Third, the set of behavioral traces is not exhaustive and may include additional indicators. In our future work, we will explore these along with a broader range of potentially coordinated IOs.

*Ethical Considerations.* To prioritize user privacy, we ensured that all control data were anonymized prior to analysis. It is crucial to note that our model's predictions might occasionally misclassify genuine accounts as coordinated, underscoring the need for careful interpretation of results. On the contrary, IO drivers mislabeled as control accounts might persist in disseminating misleading narratives or scams. As such, our model should serve as one among several tools to more accurately differentiate between IO drivers and genuine accounts. *Note: This study is IRB-approved.*

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

# APPENDIX

Figure 10 displays the Cumulative Distribution Function of four centrality measures of the co-retweet similarity network for both IO drivers and organic users from Egypt & UAE. This distribution pattern is consistent across all the countries and similarity networks we examined, leading us to choose eigenvector centrality as the selected centrality measure.

Figure 11 depicts the comparative improvements in F1 and AUC when transitioning from edge filtering to node pruning. Our findings indicate that node pruning boosts F1 and AUC by an average of 0.17 and 0.11, respectively.

Figure 12 displays the performance of the edge filtering approach with varying parameters for each behavioral trace. It's worth noting that while a specific behavioral trace might effectively detect certain IOs, it may not perform as well with others. As expected, there is a consistent trade-off between precision and recall.

Table 4 shows the classification performance of a multiscale filtering method, which does not yield enhancements in predictive performance.

Table 5 displays the classification performance of the node pruning approach for each behavioral trace and country under investigation. While it might not necessarily enhance the classification performance for every campaign, the fused approach does improve the model's generalizability across different campaigns.

Figure 13 portrays an ablation study of the supervised model based on node embedding of the fused similarity network. The results indicate that each behavioral trace contributes positively to the fused model, and removing any of them can reduce classification accuracy. Specifically, the co-Retweet and Fast Retweet similarity networks appear to be the most and least relevant inputs to the fused network, respectively.

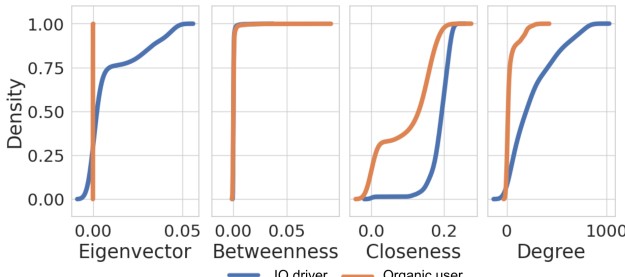

Figure 10: Cumulative Distribution Function of four network centralities (i.e., degree, eigenvector, betweenness, closeness) of a similarity network of users from Egypt & UAE.

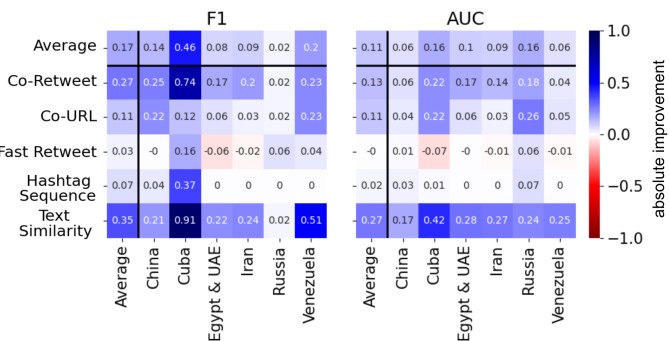

Figure 11: Improvement in AUC and F1 classification performance by using node pruning instead of edge filtering.

| Similarity Network | Recall | Precision | F1 | AUC |
|---|---|---|---|---|
| Co-Retweet | 0.47 | 0.85 | 0.54 | 0.72 |
| | ± 0.26 | ± 0.20 | ± 0.16 | ± 0.12 |
| Co-URL | 0.29 | 0.66 | 0.33 | 0.62 |
| | ± 0.38 | ± 0.33 | ± 0.37 | ± 0.15 |
| Fast Retweet | 0.22 | 0.69 | 0.27 | 0.60 |
| | ± 0.25 | ± 0.23 | ± 0.26 | ± 0.12 |
| Hashtag Sequence | 0.28 | 0.69 | 0.35 | 0.63 |
| | ± 0.27 | ± 0.29 | ± 0.26 | ± 0.13 |
| Text Similarity | 0.00 | 0.00 | 0.00 | 0.00 |
| | ± 0 | ± 0 | ± 0 | ± 0 |

Table 4: Average classification performance of backbone method.

Figure 14 illustrates the number of IO drivers who initiated their activity between 2010 and 2019.

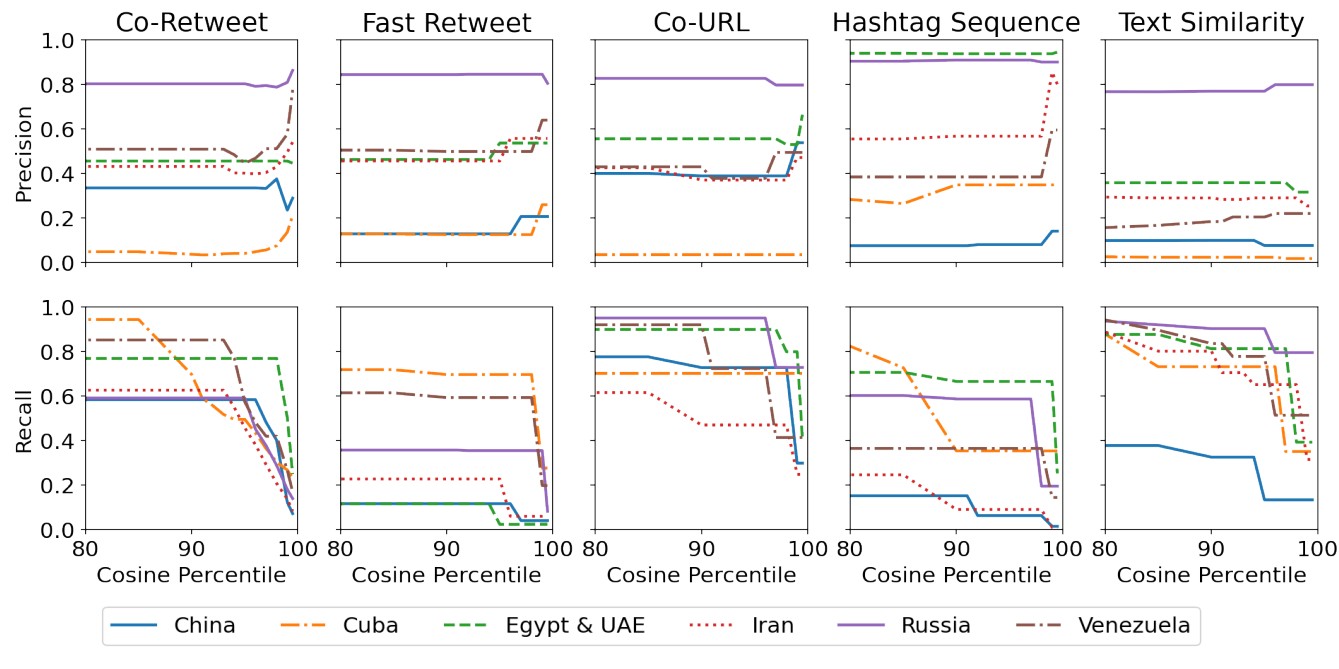

Figure 12: Precision and Recall of the edge filtering approach

| Country | FR | CR | CU | HS | TS | Fused |
|---|---|---|---|---|---|---|
| Cuba | 0.72 | 0.96 | 0.84 | 0.89 | 0.95 | **0.97** |
| Iran | 0.54 | **0.77** | 0.66 | 0.58 | **0.77** | **0.77** |
| Russia | 0.62 | 0.72 | **0.89** | 0.75 | 0.78 | **0.89** |
| China | 0.51 | 0.76 | **0.84** | 0.53 | 0.62 | 0.82 |
| Venezuela | 0.74 | 0.89 | **0.90** | 0.64 | 0.87 | 0.88 |
| Egypt & UAE | 0.52 | 0.79 | 0.80 | **0.84** | 0.78 | 0.70 |

Table 5: AUC of the node pruning approach for IOs from Egypt & UAE, Cuba, Iran, Russia, China, and Venezuela. FR = Fast Retweet, CR = Co-Retweet, CU = Co-URL, HS = Hashtag Sequence, TS = Text Similarity.

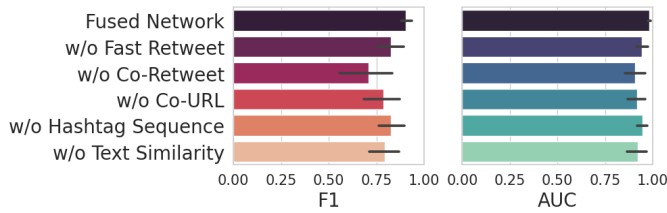

Figure 13: Ablation study: Average F1 and AUC of the supervised model based on the fused similarity network, and its possible variations

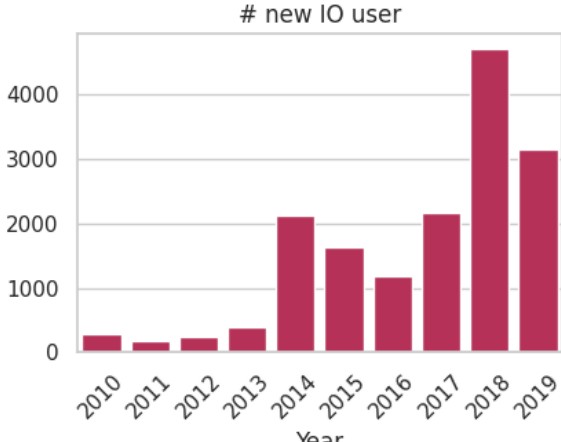

Figure 14: New active IO drivers per year

