# OpenReview forum: "Unmasking the Web of Deceit: Uncovering Coordinated Activity to Expose Information Operations on Twitter"
_ACM.org/TheWebConf/2024/Conference — TheWebConf24 Oral_

### Official Review · Reviewer_1YHs · 2023-10-28

**Novelty:** 5
**Technical Quality:** 5

**Review:**

The paper presents methods to detect coordinated activity on Twitter, using datasets released by Twitter about Information Operations and using multiple graph analysis techniques. Overall, the paper’s approach is interesting, it leverages large-scale datasets spanning multiple countries, and it proposes an interesting approach to fuse behavioral traces. The paper has the following strengths:

**Strengths:**
1. Large-scale datasets spanning multiple countries
2. The paper investigates multiple graph analysis techniques to identify coordinated activity
3. Use of five behavioral traces for the creation of the graphs and an interesting approach to fuse them

At the same time, the paper has some substantial weaknesses that are summarized below.

**Weaknesses:**
1. The control dataset does not follow the same activity patterns as the IO drives dataset
2. The paper does not explain how the optimized parameters are inferred
3. The proposed method on node centrality will likely flag influencer users as IO drivers simply because they are connected to a lot of other users

Overall, my main concern with the paper is that the control dataset has substantially different activity compared to the datasets pertaining to the IO operations. In particular, the IO operations datasets usually have fewer users, however, they have far more tweets. So it’s unclear how this discrepancy affects the results simply because the two datasets have this significant variance. The paper touches upon this in the limitations, however, there is no discussion of how this limitation likely affects the key insights from this paper. A potential way to address this issue is by subsampling the datasets in a way that two sets of IO drivers and control datasets have the same activity and repeat the same analysis.

Second, I am puzzled of how the paper extracted the optimized parameters that are mentioned in Section 5.1 I suggest clarifying how these parameters are selected and whether this parameter selection is a contribution of the paper over the existing work.

Finally, I am a little bit concerned with the proposed method in Section 5.2, given that, based on the method, they identify users that are simply connected to too many other users, which is the definition of an influencer user on social media. Influencers are very popular accounts that share content with many other organic users interacting with them simply because the account is very popular. So I am wondering how the proposed approach will distinguish between popular users (i.e., influencers) and IO drivers.

**Questions:**

1. How do you infer the optimized parameters (Section 5.1)?
2. How are the results affected because of the differences in activity between the IO drives dataset and the control dataset?
3. How does the approach proposed in Section 5.2 differ from approaches aiming to identify influencers (i.e., users that are popular and connected to many other users)?

**Ethics Review Description:**

No ethical issues

**Reviewer Confidence:**

4: The reviewer is certain that the evaluation is correct and very familiar with the relevant literature

**Scope:**

4: The work is relevant to the Web and to the track, and is of broad interest to the community

---

### Official Review · Reviewer_XtpD · 2023-11-22

**Novelty:** 6
**Technical Quality:** 6

**Review:**

This manuscript proposes an evaluation of methods for detecting influence campaigns on social media (particularly Twitter) using datasets of verified information operations in six countries. The authors test common approaches based on the thresholding of user similarity networks and develop a novel approach based on pruning nodes using their eigencentrality in the similarity network. They show that this approach usually outperforms edge filtering. They also propose a supervised machine learning approach to detect information operations combining several similarity networks. They show that their classifier performs well.
. Apart from a few needed clarifications.

*Strengths:*
- The quality, clarity, originality and significance of the work is high.

*Weaknesses:*
- About the creation of the control group. It's unclear whether collecting only 100 tweets from these users is sufficient and how it compares to the IO drivers. The authors could also have verified that these accounts did not exhibit bot-like behaviors.
- The authors show that used eigencentraliy on the similarity graph to identify IO drivers works well. But a convincing explanation of what it means for an user to have a high centrality means and why IO drivers, and not other users, should have high centrality is missing. Moreover, as the cosine similarity is not a true metric, it is not clear mathematically what this centrality measures. If the similarity between node A and B is 0.9 and between node B and C it is 0.7, it does not tell us what the similarity between node A and C is.
- More details about the structure of the similarity graphs should be given. Are they connected? dense or sparse? This would make the construction of the fused network clearer.

EDIT After authors' answers:
I am satisfied with the answers of the authors and I have updated my score accordingly. It's still not clear mathematically what is the meaning of a network built from cosine similarity, but this is not an issue for this paper and the authors showed that their approach is successful.

**Questions:**

About the control group: Why did you use only 100 tweets from the control group users? Did you verify that there were not bots? How many users do they represent in total?

About the creation of the similarity graphs: Could you add an equation describing the computation of the TF-IDF weights in terms of the network parameters? Could you give more details about the structure of these graphs?

Fig. 2: Panel B: How are these two curves computed? do you only consider edges between IO drivers for the blue one and only between organic users for the orange one? What about the edges between IO drivers and organic users?
Panel C: Why does it look like all the organic users have eigencentrality zero? And why does it look like some users have a negative centrality?

Fused network: You say that you use an unweighted version, but it's not clear how it is computed and it's not clear how the weighted version is computed either.

Fig. 8: Are the organic users also present in this figure?

**Ethics Review Description:**

No ethical issues

**Reviewer Confidence:**

3: The reviewer is confident but not certain that the evaluation is correct

**Scope:**

4: The work is relevant to the Web and to the track, and is of broad interest to the community

---

### Official Review · Reviewer_tn6o · 2023-11-24

**Novelty:** 2
**Technical Quality:** 2

**Review:**

The paper utilizes several similarity graphs to identify the state-sponsored information operations (IOs) in the social media platforms. To be specific, the paper utilizes the tweets data to construct graph based on the similarity of five features such as Co-retweet and Co-URL, then it adopts Low-weight edge filtering and network pruning to find the centralized users with high similarity weights. After that, the paper constructs a fused network by linking two nodes if they are connected in any of the individual similarity networks. The paper performs several experiments to evaluate the performance of these individual networks and the fused network.
The idea of this paper is easy to follow and the study is valuable. However, the proposed methods are trivial, only utilizing the off-the-shelf tools to build the graph without any modification based on the target task is not enough. Moreover, in the experiments, different IOs have different behaviors, such a simple fusion strategy may not eliminate the bad influence from other individual similarity networks. Finally, the annotation mechanisms of Twitter datasets are not clear, which means the data used in this study is not reliable, thus I think this study is not reliable enough.

**Questions:**

1) The research relies on the assumption that the genuine users operate independently, exhibiting limited similarities in their online behaviors. Is it really hold in social media platforms? People with similar opinions could have similar online behaviors, which is also very common.
2) How to identify the real users and IOs if they have similar behaviors just because the users agree with the opinions shared by the IOs?
3) The annotation mechanisms of Twitter datasets are not clear, there could be many wrong annotations. How to make sure the study is helpful in real-word IO detection task? The good performance could be that the Twitter just utilizes the same annotation methods.

**Ethics Review Description:**

There are no ethical issues with this paper.

**Reviewer Confidence:**

4: The reviewer is certain that the evaluation is correct and very familiar with the relevant literature

**Scope:**

3: The work is somewhat relevant to the Web and to the track, and is of narrow interest to a sub-community

---

### Official Review · Reviewer_ABVZ · 2023-11-24

**Novelty:** 5
**Technical Quality:** 5

**Review:**

Quality: The quality of this work is high. The authors have conducted a thorough analysis of coordinated activity on Twitter and have proposed a framework based on similarity graphs and network filtering techniques to accurately identify key players driving IOs. The study is well-designed, and the methodology is sound. The authors have used a large dataset of tweets and have employed various statistical techniques to analyze the data. The results are presented clearly and are supported by appropriate evidence.

Clarity: The clarity of this work is excellent. The authors have presented their ideas and findings in a clear and concise manner. The paper is well-structured, and the sections flow logically. The authors have used appropriate terminology and have explained their methods and results in detail. The figures and tables are well-designed and are easy to understand.

Originality: The originality of this work is high. The authors have proposed a novel framework for identifying coordinated activity on Twitter. The authors have also used a large dataset of tweets, which adds to the originality of the study.

Significance: The significance of this work is high. The study has important implications for understanding coordinated activity on Twitter and for identifying key players driving IOs. The proposed framework can be used by researchers and policymakers to identify and counter disinformation campaigns on social media. The study also contributes to the development of new methods for analyzing social media data.

Pros:
- Thorough analysis of coordinated activity on Twitter
- Novel framework for identifying key players driving IOs
- Large dataset of tweets
- Clear and concise presentation of ideas and findings
- Important implications for understanding and countering disinformation campaigns on social media

Cons:
- The study is limited to Twitter and may not be applicable to other social media platforms
- The study focuses on IOs and may not be applicable to other types of coordinated activity on social media

**Questions:**

- Have you considered applying your framework to other social media platforms, such as Facebook or Instagram?
- How do you plan to address the limitations of your study, such as the focus on IOs and the limited applicability to other social media platforms?
- Have you considered the ethical implications of identifying and countering disinformation campaigns on social media? How do you plan to address these ethical concerns?

**Reviewer Confidence:**

3: The reviewer is confident but not certain that the evaluation is correct

**Scope:**

3: The work is somewhat relevant to the Web and to the track, and is of narrow interest to a sub-community

---

### Official Review · Reviewer_s64C · 2023-11-30

**Novelty:** 6
**Technical Quality:** 6

**Review:**

STRENGTHS

Looks at a problem that is unfortunately still relevant to society.

Looks at the problem of classification with an interesting twist that performs well.

WEAKNESSES

The datasets used have been well studied.

It is unclear how well the methods generalize to other platforms besides Twitter.

SUMMARY

This paper is focused on the problem of detecting organized influence campaigns. Using several ground truth datasets released by Twitter over the years, the authors develop a graph based approach that exposes entire influence operation networks.

The classifier works by eschewing edge filtering techniques and instead using node pruning. Then, embeddings are extracted from the different similarity networks which are subsequently used to build a classifier which has relatively decent performance.

REVIEW

The paper is well written and influence campaigns remain a big problem that we are still struggling to address. This paper does make contributions in that direction.

My major issues with the paper are that it does not seem to be a massive advance in the state-of-the-art. Table 2, while definitely indicating an improvement on state-of-the-art, is not amazing, especially considering the +/- differences. In many cases, the prior work has much tighter intervals and the intervals for the current work overlap quite a bit with those prior work intervals.

I'm also left questioning the generalizability of this method. Twitter has certain explicit affordances that intuitively map to a graph, but those affordances don't exist (or have their own twists) on other platforms. E.g., Reddit or YouTube.

Finally, the datasets used are not particularly novel and have been well explored in the literature, but this previously exploration isn't really mentioned. For example, Savvas Zannettou did a bunch of work on the earliest official released Twitter datasets, and while the current paper is definitely not a rehash of those earlier papers, they should probably be acknowledge considering they performed graph analysis, etc.

Overall, while I think that this paper is well enough executed, I don't think it makes enough of a contribution to recommend an accept.

**Questions:**

Can you give at least some intuition as to how generalizable your technique would be outside of Twitter?

More specifically, is there any reason to believe this technique would work on platforms with different affordances? (Reddit is a good example of a platform with different affordances).

Can you elaborate on the qualitative effects of the difference in performance on your model vs. edge filtering models? The numbers indicate (somewhat) better performance, but it's not drastic and thus I'd like to know how it plays out in practice.

**Reviewer Confidence:**

4: The reviewer is certain that the evaluation is correct and very familiar with the relevant literature

**Scope:**

4: The work is relevant to the Web and to the track, and is of broad interest to the community

---

### Decision · Program_Chairs · 2024-01-22

**Decision:**

Accept (Oral)

**Comment:**

The reviewers all found the methods presented in this work to be a novel and effective approach for an important practical problem. The reviewers did point out a number of concerns, but the authors' responses were quite helpful in addressing those concerns. Given this, we're happy to recommend acceptance of this work.